# An Approach for Preliminary Landslide Scarp Assessment with Genetic Algorithm (GA)

**Chih-Ling Wang, Chi-Jyun Ko [†], Hock-Kiet Wong, Pei-Hsin Pai and Yih-Chin Tai ***

Department of Hydraulic and Ocean Engineering, National Cheng Kung University, Tainan 70101, Taiwan;
monkey93135@gmail.com (C.-L.W.); ws612531@gmail.com (C.-J.K.); ikhowong93@hotmail.my (H.-K.W.);
paipei14916@smail.nchu.edu.tw (P.-H.P.)
* Correspondence: yctai@ncku.edu.tw
† Current address: Institute of Applied Mechanics, National Taiwan University, Taipei 10617, Taiwan.

**Abstract:** For the investigation of landslide mass movement scenarios through numerical simulation, a well-defined released mass and a precise initial source area are required as prerequisites. In the present study, we present a genetic algorithm-based approach for preliminarily assessing the landslide scarp when the local field data are limited, using an ellipse-referenced idealized curved surface (ER-ICS)—a smooth surface constructed with respect to an ellipse. According to a specified depth at the center, there are two distinct curvatures along the major and minor axes, respectively. To search for the most appropriate ICS, the reference ellipse is translated, rotated, and/or side-tilted to achieve the optimal orientation for meeting the best fitness to the assigned condition (delineated area or failure depths). The GA approach may significantly enhance the efficiency, by reducing the number of candidate ICSs and notably relaxing the searching ranges. The proposed GA-ER-ICS method is examined and shown to be feasible, by mimicking the source area of a historical landslide event and through application to a landslide-prone site. In addition to evaluating the fitness of the ICS-covered area with respect to the source scarp, the impacts of various ICSs on the flow paths are investigated as well.

**Keywords:** genetic algorithm (GA); landslide-prone area; landslide scarp assessment; ellipse-referenced idealized curved surface (ER-ICS); flow paths; scenario investigation

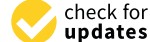



## 1. Introduction

From the viewpoint of hazard assessment or disaster mitigation, the core concerns are the plausible threats to residents and potential damage to infrastructure. Over the years, researchers have proposed various empirical laws (e.g., [1–3]) and physics-based numerical simulation tools (e.g., [4–7]) in order to delineate landslide-susceptible hazard zones. With either of these approaches, the volume of the released mass and the location of the possible failure surface are prerequisites for the evaluation process. However, predicting the released landslide volume and estimating the failure surface are highly challenging, due to high uncertainty caused by the complexity of spatial geological and hydrological variations. At the same time, weathering effects and the material composition at sites are also generally inhomogeneous (e.g., [8,9]).

With the rapid development of UAVs and modern remote sensing techniques (e.g., LiDAR, UAV-LiDAR, SAR, InSAR, and UAV-SAR), high-resolution digital elevation models (DEMs) have become popular, in which detailed topographic features can be well-recognized. Furthermore, with expensive and time-consuming geological field surveys, the scar boundaries of landslide-prone areas can be estimated and delineated, with respect to some specific features (e.g., crowns, bulges, trenches, or fissures). Based on the DEM, a geometric interpretation method—namely, the Sloping Local Base Level (SLBL) method—has been suggested by Jaboyedoff et al. [10,11] for the approximation of a 3D

failure surface, which possesses a constant second derivative in the down-slope direction (i.e., a parabolic curve in section view). Besides the SLBL approach, Reid et al. [12,13] have proposed a spherical surface for analyzing the slope stability of a 3D stratovolcano edifice. Considering the spatial geological structure and groundwater patterns, the concept of spherical failure surfaces has been adopted and extended in the open-source software Scoops3D (see, e.g., [14,15]). Tun et al. [16] have further applied Scoops3D for calculating the probability of multiple failures, where a genetic algorithm (GA) with the first-order reliability method (FORM) was utilized.

For a well-defined landslide scarp area, Kuo et al. [17] have proposed a smooth minimal surface (SMS) method to approximate the failure surface by a smooth surface, where the constructed fracture surface is determined according to the minimal surface area, with the prerequisites of a given landslide volume and a convex polygon-outlined region. Instead of fitting the scarp boundary, Tai et al. [18] suggested the concept of using an idealized curved surface (ICS) to mimic the fracture surface for numerical simulation, where the ICS is defined by two distinct curvatures in the down-slope and cross-slope directions, respectively (cf. Figure 1b). As the ICS approach does not request an exact fitness to the assigned area (source area or delineated area), a search process is needed to find the most appropriate ICS. Motivated by the ellipticity of landslide shapes [19], and for ease of proceeding with the search process, Ko et al. [20] have utilized an ellipse with a specified depth to represent the corresponding ICS (ellipse-reference ICS; ER-ICS), where the depth is used to determine the landslide volume. In the search process, the optimal reference ellipse is selected, with respect to the best fitness to the assigned area, through translation, rotation, and/or side-tilting (cf. Figure 2). The ER-ICS search process is an exhaustive method considering all the candidate ICSs, where the associated depth is determined by the assigned volume. Due to the non-trivial topography, the reference ellipse-covered area varies for different orientations, such that the determination of depth should be repeated for each candidate ICS. Therefore, the search process is rather time-consuming, limiting the search range. Taking the ICS-D in Ko et al. [20] as an example, the search range covers $7 \times 7$ grids for the top and bottom vertices, yielding 10,633 candidate ellipses with corresponding ICSs to be evaluated. Hence, an efficient method, which either reduces the number of candidate ellipses or enhances the computational performance, is highly desirable. Accordingly, a genetic algorithm (GA) [21–24] can provide an optimal solution for reducing the number of candidate ellipses.

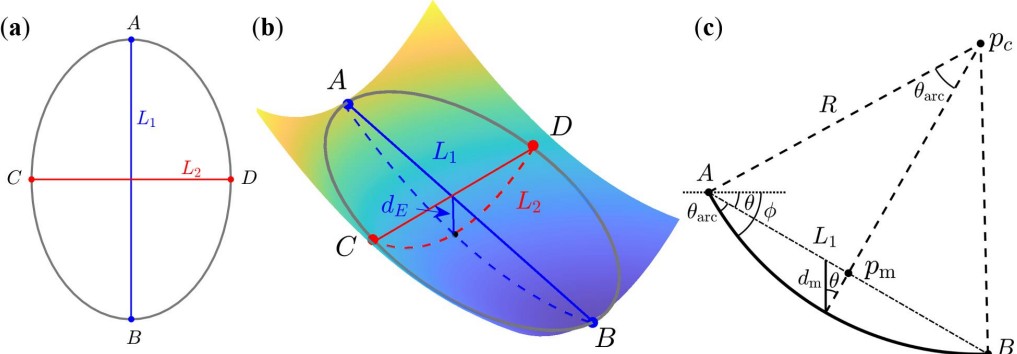

**Figure 1.** Ellipse-referenced idealized curved surface (ER-ICS): (**a**) The reference ellipse with major axis $L_1$ and minor axis $L_2$; (**b**) The constructed ICS with respect to a depth $d_E$ below the middle point; (**c**) Section view of the ICS along the major axis, where $d_m \cos \theta = d_E$ with inclination angle $\theta$.

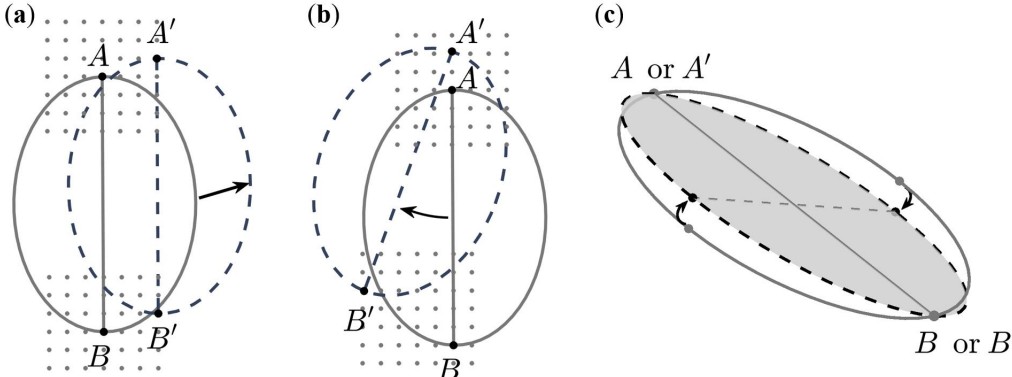

**Figure 2.** To search for the optimal ellipse, the reference ellipse is (**a**) translated, (**b**) rotated, and/or (**c**) side-tilted.

From the viewpoint of engineering applications, a genetic algorithm (GA) is an optimization tool utilizing an iterative process. It delivers solutions through the use of evolution operators such as mutation, crossover, and selection [24]. In the present study, the "canonical genetic algorithm" (cf. [21,23,24]) is adopted for the search process; that is, the initial population is generated with the first reference ellipse mapping within the assigned search ranges of parameters randomly. The parameters determining the orientation of the reference ellipse comprise the genes. Each constructed ICS is evaluated based on the fitness function, and then assigned a fitness value. The selection process follows the method of *roulette wheel selection* [24], where the possibility of being selected is proportional to fitness (also known as *fitness proportionate selection*). After the selection process, recombination (crossover) of the genes (parameters determining the orientation of the reference ellipse) is conducted for breeding the populations of the next generation, where mutation is allowed with a slight possibility (generally ≤1%).

In addition to the GA approach, a manipulation treatment is introduced to isolate some redundant portion(s) of the ICS-covered area, as the complex topography may induce the inclusion of unexpected regions in the neighborhood of the target area. The redundant part can be automatically isolated without additional manual operation, through utilization of *Erosion* and *dilation* operations in morphological image processing (e.g., [25]). This manipulation treatment may retain an ICS whose main portion has good fitness, while removing the redundant part(s).

The feasibility and applicability of the GA-ER-ICS approach are investigated through the validation of a historical landslide event and application to a plausible failure surface, based on the measured failure depths in a landslide-prone area. In terms of the fitness of the target area, the convergence of the employed GA approach is examined, considering the number of generations. The application to landslide-prone areas demonstrates another advantage of the GA-ER-ICS approach, in that the failure surface can be mimicked in a flexible manner for various scenarios. All of the selected ICSs with the corresponding released volumes of landslide mass are integrated into a GPU-accelerated simulation tool (MoSES_2PDF [26]), in order to investigate the impacts of various ICSs on their consequent flow paths.

The remainder of this manuscript is structured as follows. In Section 2, the construction of ellipse-referenced ICS is reviewed, where the manipulation approach for isolating the redundant portion of the ICS-covered area is detailed. In Section 3, the employment of the GA procedure is elaborated. The procedure of the GA-ER-ICS searching process is given in Section 4. Numerical investigations and the application to a landslide-prone area are discussed in Section 5. The key features of the proposed GS-ER-ICS approach and its potential for engineering applications are summarized and highlighted in the concluding remarks.

## 2. Ellipse-Referenced Idealized Curved Surface (ER-ICS)

### 2.1. Construction of the ER-ICS

The ICS concept was first suggested to replicate the failure surface of a sliding-type landslide in Tai et al. [18]. As the ICS is characterized by two constant curvatures in the down- and cross-slope directions, respectively, Ko et al. [20] used an ellipse with a specified depth to construct the ICS; that is, the Ellipse-Referenced Idealized Curved Surface (ER-ICS). As shown in Figure 1a, $\overline{AB}$ (with length $L_1$) denotes the major axis of the ellipse along the down-slope direction, while $\overline{CD}$ (with length $L_2$) represents the minor axis. Once $L_1$ and $L_2$ are fixed, the ICS can then be constructed with respect to a specified depth, $d_E$, where the corresponding curvature radii ($R_1$ and $R_2$) are determined accordingly (cf. panels b and c). In general, point $A$ sits at the upper part of the scarp area, point $B$ is at the lower part, and the length $L_2$ of $\overline{CD}$ determines the width of the site. Once the ICS is constructed, the landslide volume can be calculated based on the DEM; that is, the reference ellipse defines the orientation and location of the ICS, while the depth determines the two curvatures and the associated landslide volume.

For each reference ellipse, the corresponding depth is determined in accordance with an assigned prerequisite, such as the released volume of mass or a specific failure depth at some location. Hence, it takes time to find the covariant depth for each reference ellipse. There may be thousands of reference ellipses for one site, where each ellipse yields an ICS and serves as a candidate. To construct the most appropriate ICS, Ko et al. [20] have suggested trying various locations/orientations of the ellipse (i.e., translating, rotating, or side-tilting the reference ellipse; cf. Figure 2), where four methods (methods A–D) were considered and evaluated. In the present study, method D in Ko et al. [20] is employed, in which, in addition to translation, rotation, and/or side-tilting, the RE is allowed to stretch or shrink slightly, while keeping the area of the RE invariant. Even though the area of the ellipse is fixed, thousands of candidate ICSs still have to be constructed. If not additionally specified, the most appropriate ICS is the one with the minimal deviation index, calculated as:

$$\Lambda_S = \frac{|A_{\mathrm{ICS}} - A_{\mathrm{ICS}\cap\mathrm{sa}}| + |A_{\mathrm{sa}} - A_{\mathrm{ICS}\cap\mathrm{sa}}|}{A_{\mathrm{sa}}}, \tag{1}$$

which indicates the deviation of the ICS-covered area from the source (target) area (cf. [20]). In (1), $A_{\mathrm{ICS}}$ denotes the ICS-covered area, $A_{\mathrm{sa}}$ is the source (target) area, and $A_{\mathrm{ICS}\cap\mathrm{sa}}$ represents their intersection.

### 2.2. Manipulation of the ER-ICS-Covered Area

Due to the complex topography, the constructed ICS might intersect the neighboring hill and include some additional unexpected area. An example can be seen in Figure 3a, in which the blue area indicates the source (target) area and the red dot line depicts the outline of the ICS-covered area. A sizable unexpected area, indexed by "II", can be seen on the right-hand side. Figure 3b shows the section view of the dashed line in panel a, where the portion in tawny color indicates the ICS-determined landslide body of interest, while the intersected part of the unexpected portion (II) is marked in yellow. This unexpected portion can be seen as redundant and is usually isolated manually (e.g., as in [20]), which is highly time-consuming and could differ between individuals. Here, a manipulation process (morphological process), on the basis of the OpenCV software [27], is introduced in order to segment and isolate the redundant section(s) automatically.

The manipulation process consists of two operations: Erosion and Dilation. The *Erosion* process shrinks the area through a local minimum over the area of a given kernel $\mathcal{K}$ (e.g., an $n \times n$ matrix or a circle). It replaces the image pixels (grid) in the anchor point (center of $\mathcal{K}$) with the local minimum. The *Dilation* process extends the area shrunk in the *Erosion* process by determining the local maximum for the anchor point over the kernel-covered area. In the present study, the kernel $\mathcal{K}$ is set using a $3 \times 3$ matrix, which is scanned over the whole DEM in the erosion and dilation operations (cf. Figure 3). Letting

the grids within the ICS-covered area be indexed by 1 and 0 set for those in the rest of the area (i.e., as a mask), one can divide the connected areas into several blocks by carrying out the Erosion operation several times, in order to isolate the redundant one(s). After that, the resultant mask for the major scarp area is retained and recovered through several runs of the Dilation operation. Figure 3c depicts the eroded area(s) after five operations, where the redundant portion (II) is separated from the primary area (I). The recovered major scarp area is illustrated in Figure 3d, where six Dilation operations have been conducted. It should be noted that the resultant scarp area is valid only when the ICS sits below the topographic surface locally. The additional Dilation operation (versus the five Erosion operations) is conducted to alleviate the cliffs possibly sitting at the boundary of the resultant mask, if they exist.

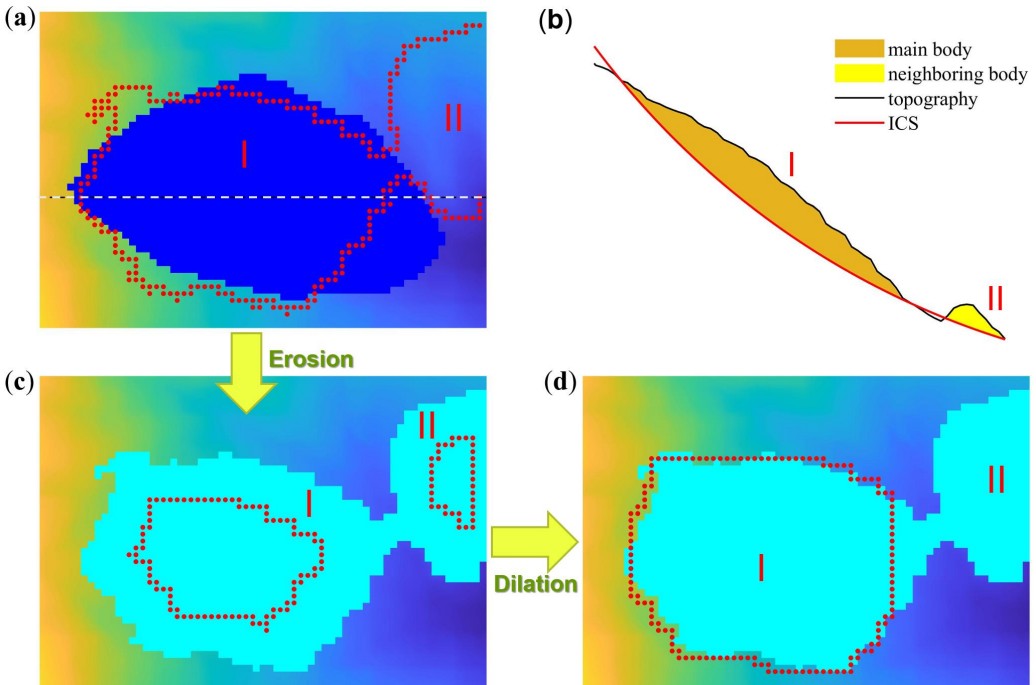

**Figure 3.** Manipulation of the ICS-covered area. The blue area represents the source (target) area, and the preliminary ICS-covered area is shown in aqua-blue. (**a**) The initial ICS-covered area is marked by the red dotted line and divided into two portions (I and II); (**b**) Section view along the white dashed line in panel (**a**); (**c**) Outlines of the two portions after Erosion operations; (**d**) The resultant mask of the target portion after Dilation operations.

## 3. Genetic Algorithm Approach

For high efficiency in the search process, a genetic algorithm (GA) is employed (see, for example, [22–24,28]). Together with the GA approach, determination of the failure depth (i.e., $d_E$, as indicated in Figure 1b) is implemented based on the CUDA structure (cf. [29]) for high-performance GPU computation. A GA is a search process, which is generally used to generate an optimal solution under a given context. The approach used here is composed of two parts: the first consists of decoding the genes which determine the candidate solutions (i.e., the orientation of the reference ellipse; RE), while the second part involves the fitness function, which is used to evaluate the corresponding performance. In this study, the genes used for constructing the RE are $(\delta x, \delta y, \delta\theta, \delta L_1)$, where $\delta x$ and $\delta y$ denote translations, $\delta\theta$ denotes rotation, and $\delta L_1$ represents stretching/shrinking of the reference ellipse. Each reference ellipse is associated with a depth, in order to construct the ICS and meet the assigned landslide volume. If not additionally specified, the deviation index, $\Lambda_S$—defined in (1)—represents the fitness (i.e., serves as a fitting function) between the ICS-covered area and the target area.

In the GA search process, thirty ICSs (population size) are constructed in each generation. During each successive generation, the better-fitted ICSs (i.e., those with smaller deviation index) are selected to breed the next generation through genetic crossover and mutation. The DNA size is 12, and the rates for crossover and mutation are 0.8 and 0.01, respectively. Figure 4 illustrates the framework of the GA-ER-ICS search process. Each ICS is constructed in the GPU section, where the depth is determined based on the assigned volume and the given genes for the orientation of the RE. The GA procedure in the CPU section evaluates the fitness and provides the genes through crossover and mutation operations. We experimentally determined that a plateau of fitness was reached after circa 10 generations. As this result might depend on the size of the search range of genes, we set 10 and 15 generations as the termination criteria for GA-ER-ICS construction in the following numerical investigation and site application, respectively.

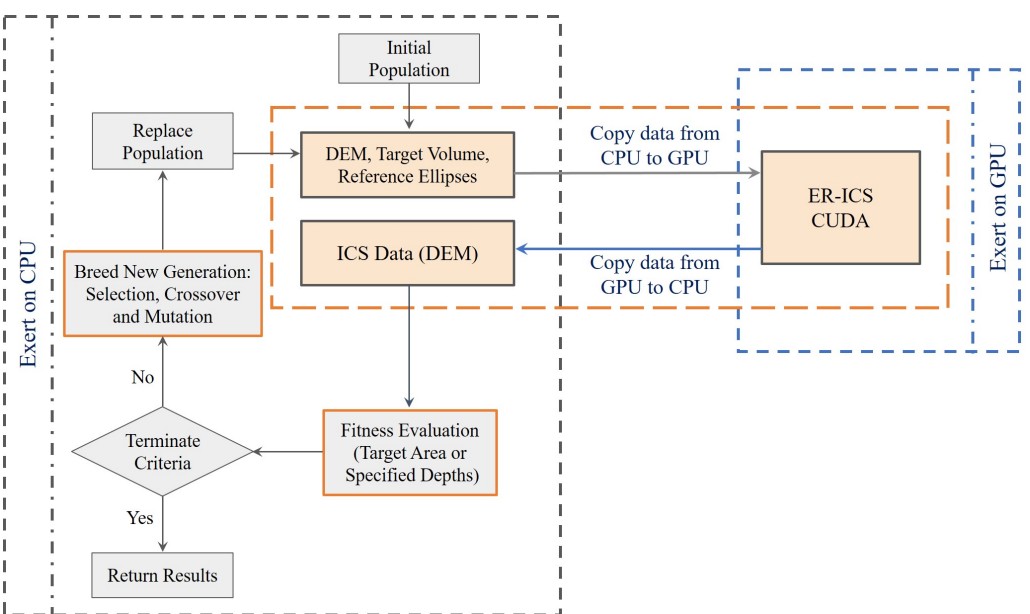

**Figure 4.** The framework of the GA-ER-ICS search process, in which the ICSs are constructed in the GPU section and the GA is operated in the CPU section.

## 4. Procedure of the GA-ER-ICS Search Process

The search process consists of four main stages: (1) Preparation; (2) Input and parameter setting; (3) ER-ICS construction; and (4) GA process, termination, and results output.

Stage 1: Preparation and determination of the initial reference ellipse

Based on the source area (post event) or the delineated area of the landslide-susceptible zone, four starting reference points are assigned to determine the initial reference ellipse (RE) on the DEM (cf. Figure 5a). In general, these four reference points indicate the length and width of the target area. The highest and lowest points (in elevation) compose the major axis of the initial RE, while the length of the minor axis is determined by regression for a minimal root mean square (RMS), with respect to the other two reference points.

Stage 2: Input and parameter setting

In this stage, the ranges of the genes $(\delta x, \delta y, \delta \theta, \delta L_1)$ are given, and the size of the population in each generation, as well as the termination criterion (the number of generations), must be assigned. With the initial RE determined in Stage 1, the initial population is randomly generated within the range of the genes.

Stage 3: ER-ICS construction

For each RE in the initial population (or the replaced population of the evolution), the individual depth is determined according to the assigned released volume, which can be

given by assignment or through the use of a volume–area empirical relation, as suggested in, for example, [17,18,30]. As each RE is associated with one most appropriate ICS, this process (including the manipulation process introduced in Section 2.2) is highly time-consuming. It is, therefore, conducted in the GPU section (cf. Figure 4). Once all the ICSs are constructed, the GPU section returns the corresponding sets of genes and DEMs to the CPU section.

Stage 4: GA process, termination, and results output

The fitness of all determined ICS (DEMs) is evaluated based on the deviation index $\Lambda_S$ given in (1), in the case where the source/target area is available. On the other hand, when the evaluation is based on the specified failure depth(s), the fitness is computed based on the root mean square (RMS) between the ICS and the specified depth(s). In both cases, a smaller value of the fitness index (i.e., $\Lambda_S$ or RMS) indicates better fitness. As illustrated in Figure 4, the next generation is bred as the new population, until the termination criterion (i.e., the maximum number of generations) is fulfilled. The best-fitted ICS in the last generation is considered the most appropriate ICS in the GA search process.

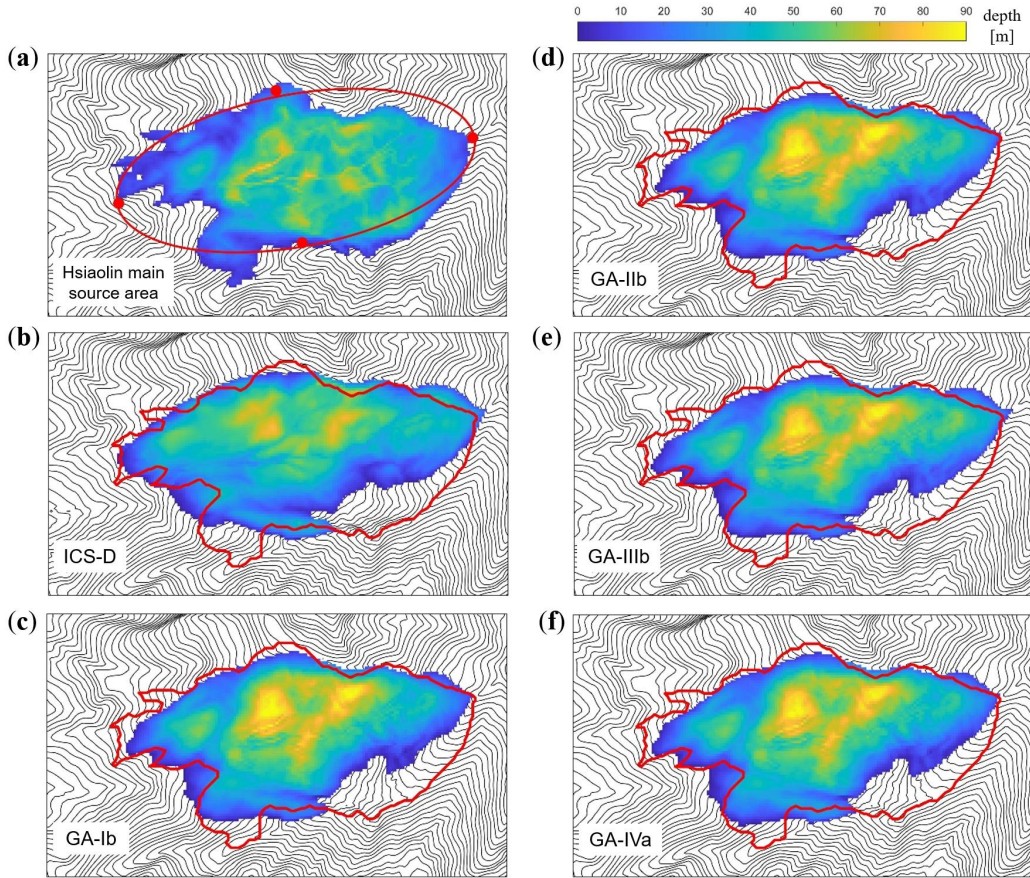

**Figure 5.** The shape and depth distribution of the initial landslide mass: (**a**) The measured data in the main source area, where the four red points stand for the starting reference points and the red line represents the initial reference ellipse; (**b**) The ICS constructed by the method-D in [20]; (**c**) The best-fitted ICS in condition GA-I (i.e., GA-Ib); (**d**) The best-fitted ICS in condition GA-II (i.e., GA-IIb); (**e**) The best-fitted ICS in condition GA-III (i.e., GA-IIIb); (**f**) The best-fitted ICS in condition GA-IV (i.e., GA-IVa).

## 5. Numerical Investigation and Application in a Landslide-Prone Area

The performance of the proposed GA-based ER-ICS (GA-ER-ICS) method, in terms of constructing a plausible failure surface, was investigated against a historical large-scale landslide event, and a trial application was arranged in a landslide-prone area. The

historical event was the Hsiaolin landslide, triggered during typhoon Moratok in 2009 in southern Taiwan. Due to its huge scale, severe damage (more than 450 victims), and unique characteristics, the 2009 Hsiaolin landslide has become one of the most representative deep-seated landslides worldwide, for which ample well-documented data are available. For further details, the readers are referred to [31–37]. The DEM and initial conditions used in [32,38] were employed in the numerical investigation. Note that only the main source area listed in Kuo et al. [32] was taken into account when mimicking the failure surface, as it provides more than 94% of the total landslide mass.

For application of the GA-ER-ICS method to a landslide-prone area, we assigned a potential large-scale landslide area named Kuanghua-T002, located in Taoyuan county, northern Taiwan. As creeping movements have been detected in this area for years, three inclinometers (indexed by K18-1BW, K18-2BW, and K19-1BW) had been installed. Both K18-1BW and K18-2BW were broken on 31 May 2018, while K19-1BW was broken on 3 December 2019, due to local mass movements (cf. Chen [39]). The plausible outline of the potential deep-seated landslide has been suggested and delineated in Chen [39].

The ranges of genes used in the GA procedure to search for the most appropriate ICSs are listed in Table 1. Due to the complex topography in the neighborhood of the target area, redundant areas were found and, so, the manipulation process was employed for site Kuanghua-T002. For reference landslide volumes, we took the measured amount (21,180,535 m$^3$) in the Hsiaolin case and 348,581 m$^3$ (estimated by the volume–area empirical relation suggested in [30]) for the Kuanghua-T002 case. The consequent flow paths of the released landslide masses were computed using a GPU-accelerated simulation tool (MoSES_2PDF in [26]). The values of the material parameters used for computation were identical to those presented in [20], as collected in Table A1. The following index:

$$\Lambda_{\text{path}} = \frac{|A_\alpha^{\text{path}} - A_{\alpha \cap \beta}^{\text{path}}| + |A_\beta^{\text{path}} - A_{\alpha \cap \beta}^{\text{path}}|}{A_{\alpha \cap \beta}^{\text{path}}}, \tag{2}$$

was introduced for quantitative investigation of the discrepancy of flow paths between two scenarios. In (2), $A_{\alpha \cap \beta}^{\text{path}}$ denotes the intersection (overlapped area) of the flow paths between Scenarios $\alpha$ and $\beta$. In the scenario campaigns, the index $\Lambda_S$ quantifies the fitness of the constructed ICS, while $\Lambda_{\text{path}}$ evaluates the difference between two ICSs (scenarios).

**Table 1.** Range of genes used in the GA for plausible ICS searches in the 2009 Hsiaolin event and site Kuanghua-T002 cases.

|  | $(\delta x,$ | $\delta y,$ | $\delta \theta,$ | $\delta L_1)$ | **Manipulation** | **Volume (m$^3$)** |
|---|---|---|---|---|---|---|
| Hsiaolin event | $(\pm 8 \Delta x,$ | $\pm 8 \Delta y,$ | $\pm 15°,$ | $\pm 8 \Delta x)$ | no | 21,180,535 |
| Kuanghua-T002 | $(\pm 5 \Delta x,$ | $\pm 5 \Delta y,$ | $\pm 10°,$ | $\pm 5 \Delta x)$ | yes | 348,581 |

*5.1. Numerical Investigation (The 2009 Hsiaolin Landslide)*

With respect to the main source area of the 2009 Hsiaolin landslide, the most appropriate ICS was selected through the GA-ER-ICS search process. Here, the feasibility and applicability are investigated in terms of three aspects: (a) Convergence with respect to the number of generations; (b) the effectiveness of the side-tilting operation; and (c) the impacts of the different coverages between ICSs on the consequent flow paths. The resolution of the used digital elevation map (DEM) was 10 m ($\Delta x = \Delta y = 10$ m), and the computational domain was $3700 \times 2210$ m$^2$, where the projection of the main scarp on the horizontal plane covered an area of 624,900 m$^2$.

Four conditions for the GA-ER-ICS search process were arranged, and three runs for each situation were carried out (cf. Table 2). The impact of the number of generations (i.e., 10 or 15 generations) on the GA process was also investigated. The search processes with side-tilting are indexed by GA-I and GA-II, in which the tilting angle was determined by

the inclination of the minor axis of the reference ellipse to the horizon (i.e., the inclination angle of $\overline{CD}$, shown in Figure 1b). GA-I and GA-III used ten generations, while GA-II and GA-IV used 15 generations. The constructed ICSs were compared with the main source area. In addition, the most appropriate ICS (ICS-D) in Ko et al. [20], as determined by method D, was included in the campaigns for comparison. We first focused on the fitness of the landslide source area and the sensitivity of the ICSs constructed by the proposed GA-ER-ICS approach under various conditions. After that, the consequent flow paths were computed, with respect to the best-fitted ICSs. The flow paths were determined using the moving mass-covered areas of 61 sets of results from 0.0 s to 181.83 s with an interval of 3.0305 s, where only areas with flow thickness greater than 10 cm were considered.

**Table 2.** Fitness of the ICSs under various conditions for the main source area in the 2009 Hsiaolin event.

| Condition | Run | Generation | Side-Tilting | $\Lambda_S$ |
|---|---|---|---|---|
| ICS-D | | - | - | 28.72% |
| GA-I | a | 10 | yes | 30.05% |
| | b * | 10 | yes | 25.01% |
| | c | 10 | yes | 25.83% |
| GA-II | a | 15 | yes | 24.08% |
| | b * | 15 | yes | 24.05% |
| | c | 15 | yes | 24.24% |
| GA-III | a | 10 | no | 25.76% |
| | b * | 10 | no | 24.23% |
| | c | 10 | no | 29.12% |
| GA-IV | a * | 15 | no | 24.15% |
| | b | 15 | no | 25.52% |
| | c | 15 | no | 24.82% |

* The best-fitted among the three runs.

5.1.1. Fitness to the Main Source Area

With respect to the starting reference points (the four red points marked in Figure 5a), three runs were performed for each condition (GA-I to GA-IV). The performance of the selected ICS was evaluated with respect to the fit to the measured source area, as indexed by the value of $\Lambda_S$ defined in (1); see Table 2. A smaller value of $\Lambda_S$ indicates a better fit. As the GA process does not guarantee identical results, the $\Lambda_S$ value varied with each run. It was found that ten generations (GA-I and GA-III) did not provide satisfactory convergence of fitness; however, a plateau of fitness is reached when the evolution terminated at 15 generations. Despite the small difference, inclusion of the side-tilting operation slightly improved the fitness (see GA-I vs. GA-III and GA-II vs. GA-IV). Furthermore, it was found that 10 of the resultant ICSs, among the 12 runs of the GA search process, presented better fitness than ICS-D. The better performance of the GA approaches is possibly due to the more considerable translation distance (i.e., ±8 grids in the GA approach versus ±3 grids in the method of exhaustion). Together with its high efficiency, the proposed GA approach is apparently superior to the exhaustive method.

The best-fitted ICSs in the four conditions (GA-Ib, GA-IIb, GA-IIIb, and GA-IVa) are exhibited in Figure 5. The measured depth distribution of the released landslide mass from the primary source area is shown in panel a, and its outline is depicted in all the other panels for comparison. Panel b displays the ICS-D result. All the ICSs, including ICS-D, yielded higher thickness than the measured data (cf. Figure 5). This phenomenon is suspected to have been induced by two causes. The first is the smaller ICS-covered area, such that a higher thickness of the initial landslide body is required to retain the constant reference volume. The second reason could be that the ICS is a smooth surface, differing from the natural failure surface, for which the local geological conditions may play a crucial

role. Another interesting finding is that the GA-selected ICSs were rather similar, for which the value of $\Lambda_S$ ranged from 24.05% (GA-IIb) to 25.01% (GA-Ib). It was also found that the side-tilting operation did not significantly improve the performance with either 10 or 15 generations, if there were three runs for selection in each condition. Hence, the operation of side-tilting was not employed for the application in a landslide-prone area (Section 5.2).

5.1.2. Impacts on Flow Paths

Ko et al. [20] have investigated the consequent flow paths of the ICS by comparing the results computed with ICS-D against the results calculated using the measured landslide scarp. The area discrepancy between the measured failure surface (FS) and ICS-D was 26.82% at the initial stage of initiation, which reduced to circa 9.5% for the computed results at the rest state. Here, we focus on the discrepancy of the flow paths computed with the selected ICSs. The differences between the initial areas and the discrepancy of the flow paths are quantitatively denoted by the indices $\Lambda_{path}^{init}$ and $\Lambda_{path}$, respectively (see Table 3). In agreement with the illustration in Figure 5, where all the GA process-selected ICSs can be seen to be rather similar, the values of $\Lambda_{path}^{init}$ were in the range from 4.13% to 6.90% (cf. Table 3). With 15 generations, no significant discrepancy between GA-IIb (with side-tilting) and GA-IVa (without side-tilting) could be identified, with respect to the initial ICS-covered area and the consequent flow paths, where $\Lambda_{path}^{init} = 4.13\%$ and $\Lambda_{path} = 3.87\%$.

**Table 3.** Comparison of the flow paths under various conditions for the 2009 Hsiaolin event.

| Impacts | ICS | $\Lambda_{path}^{init}$ | $\Lambda_{path}$ |
|---|---|---|---|
| side-tilting | GA-Ib vs. GA-IIIb | 6.90% | 10.52% |
| | GA-IIb vs. GA-IVa | 4.13% | 3.87% |
| generation | GA-Ib vs. GA-IIb | 5.38% | 6.06% |
| | GA-IIIb vs. GA-IVa | 5.70% | 7.77% |
| Exhaustion vs. GA | ICS-D vs. GA-Ib | 15.08% | 14.24% |
| | ICS-D vs. GA-IIb | 14.77% | 12.63% |
| | ICS-D vs. GA-IIIb | 12.84% | 13.97% |
| | ICS-D vs. GA-IVa | 14.53% | 13.13% |

Among the four selected best-fitting ICSs (GA-Ib, GA-IIb, GA-IIIb, and GA-IVa), the results computed with GA-Ib had the most significant discrepancy from the results, compared with ICS-D, where $\Lambda_{path}^{init} = 15.08\%$ and $\Lambda_{path} = 14.24\%$. Figure 6 depicts the associated distinctions at the initial stage (panel a) and the flow paths (panel b). The four red markers in panel a are the initial reference points in the search process, and the red line outlines Hsiaolin village. It was found that most of the differences took place around the source area, due to the shape discrepancy in the initial stage. For a thorough overview, the flow paths of ICS-D, GA-Ib, GA-IIb, GA-IIIb, and GA-IVa were all collected, and are displayed in a comparable way in Figure 7, in which the blue zone represents the shared paths. The individual routes yielded by ICS-D, GA-Ib, GA-IIb, GA-IIIb, and GA-IVa are shown in red, magenta, cyan, green, and yellow, respectively. Due to the minor distinction between the source areas, only one of the two outlines is given in each comparison scenario. We refer the readers to Figure A1 in Appendix B for the details of the outlines between the source areas, in accordance with the sequence of Figure 7. As has already been elaborated and illustrated in Figure A1e, the flow paths of GA-IIb and GA-IVa were very close, with $\Lambda_{path} = 3.87\%$. Both of them were constructed using 15 generations in the GA process, revealing that, with a sufficient number of generations for evolution, the effectiveness of the side-tilting operation becomes insignificant. The data listed in Table 3 indicate that the GA-ER-ICS method may deliver the selected ICSs with good convergence, in terms of the flow paths. This convergence might benefit from the repetition of operations and a sufficient number of generations for evolution in the GA search process.

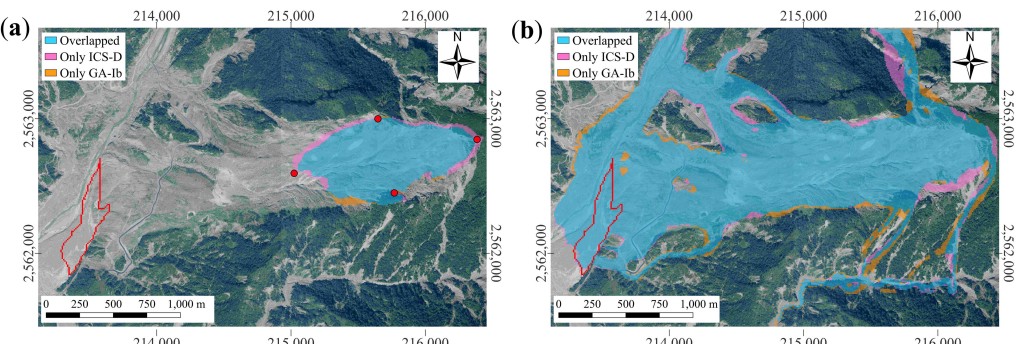

**Figure 6.** Orthophoto and landslide mass-covered areas in the computation, where cyan indicates the overlapped area, the pink area is covered only by the results computed with the ICS-D, and the area in dark-yellow is occupied only in the condition of GA-Ib. The red line denotes Hsiaolin village, and the four red markers in panel a represent the initial reference points in the GA-ER-ICS search process. (**a**) Initial stage; (**b**) Flow paths (Orthophoto: Courtesy of Serial Survey Office, Forestry Bureau, Taiwan).

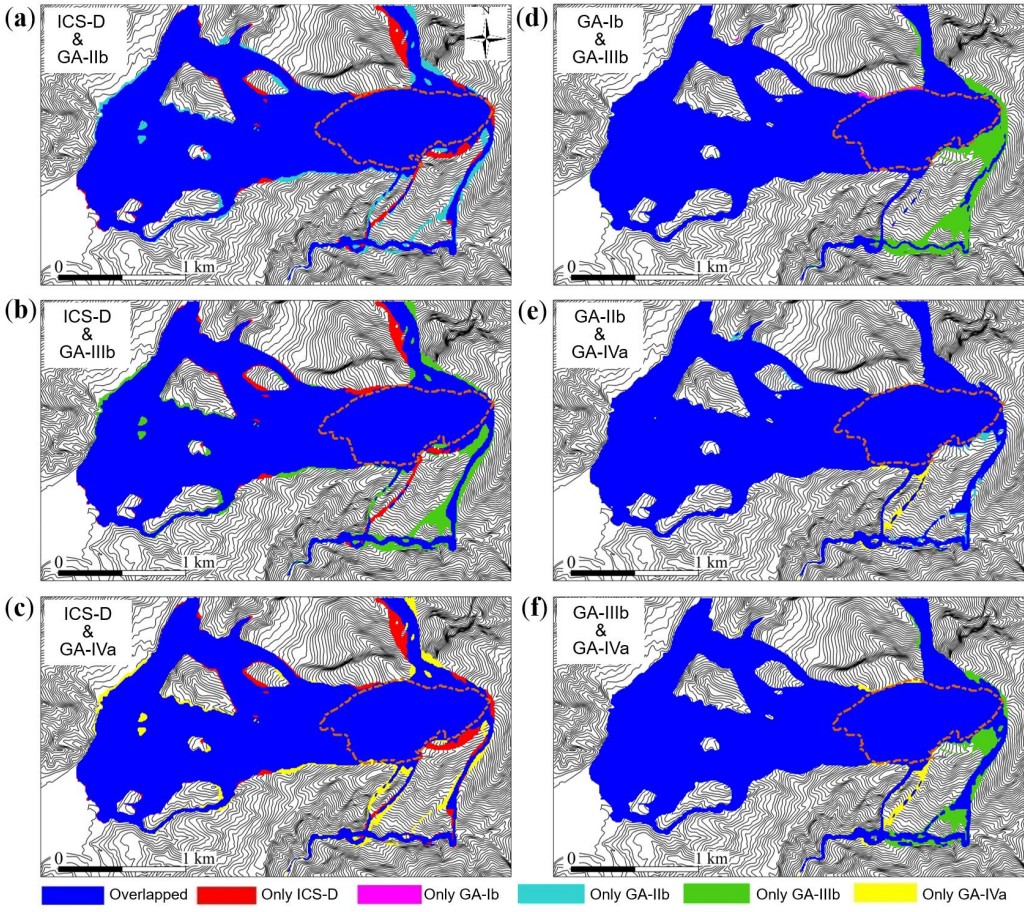

**Figure 7.** Flow paths in various conditions, where the blue represents the overlapping area. The red, magenta, cyan, green, and yellow areas are covered only by results computed with ICS-D, GA-HL-Ib, GA-HL-IIb, GA-HL-IIIb, and GA-HL-IVa, respectively. The brown dash-dotted lines outline the source areas: (**a**) ICS-D; (**b**) GA-IIIb; (**c**) GA-IVa; (**d**) GA-Ib; (**e**) GA-IIb; (**f**) GA-IVa.

*5.2. Application to a Landslide-Prone Area (Kuanghua-T002)*

Based on the records of the three installed inclinometers (K18-1BW, K18-2BW, and K19-1BW), three scenarios (A, B, and C) were designed, as listed in Table 4. In Scenario A, the fitness evaluation is based on the failure depths recorded by gaging wells K18-1BW

and K18-2BW. Only K19-1BW is considered in Scenario B, while the failure depths of all three gaging wells are taken into account in Scenario C. In this application, the DEM has a resolution of 5 m and the horizontal projection of the delineated source area is 32,325 m$^2$. This is much smaller than the main source area in the Hsiaolin case, so we set $\delta x = \delta y = \delta L_1 = \pm 5\Delta x$ in the GA-ER-ICS search process.

**Table 4.** Scenarios for the Kuanghua-T002 landslide-prone area.

| Scenario | Referenced Gaging Well | ICS-Determined Failure Depth (m) | | | $\Lambda_S$ | $V_{\text{Target}}/V_{\text{Guzzetti}}$ |
|---|---|---|---|---|---|---|
| | | **K18-1BW** | **K18-2BW** | **K19-1BW** | | |
| A | K18-1BW & K18-2BW | 5.68 | 24.98 | (38.71) | 23.36% | 1.5 |
| B | K19-1BW | (none) | (14.08) | 27.35 | 29.47% | 1.0 |
| C | K18-1BW, K18-2BW & K19-1BW | 4.63 | 19.27 | 30.71 | 36.58% | 1.0 |
| Inclinometer records | - | 5 | 25 | 27 | - | - |

Note: Values in brackets are not for the fitness evaluation in the GA search processes.

Although local movement and failures have been detected by the inclinometers, they did not take place at the same time, and the landslide body has not yet been ultimately released. Hence, the exact volume of the landslide mass is not available for determining the depth, as well as constructing the ICS. The reference volume for constructing the ICS can basically be approximated by empirical laws based on the delineated source area, such as Guzzetti's empirical volume–area relation [30] or other similar laws, as detailed in [17,18]. As the ICS-determined failure surface is required to be close to the records of the inclinometers, with as sound a fitness to the delineated area (e.g., $\Lambda_S \leq 40\%$) as possible in the GA-search process, one may not always obtain satisfactory results. In such circumstances, we can relax the constraint of the reference (target) volume; for example, the reference volume $V_{\text{Target}}$ in Scenario A was suggested as $1.5 \times V_{\text{Guzzetti}}$ for a more satisfactory result. The best-fitting ICSs in Scenarios A, B, and C are listed in Table 4, where the values in brackets were not taken into account when evaluating the fitness in the GA search processes.

The best-fitting ICSs in Scenarios A, B, and C are illustrated in the left panels of Figure 8. The yellow line depicts the outline of the delineated area, while the ICS-covered regions are marked in an emerald-green color. In Figure 8a, the four red markers represent the reference points for determining the first RE in the GA-ER-ICS search process. As reported in Table 4, the discrepancy between the ICS-determined source area and the delineated one was indexed by $\Lambda_S$, whose value ranged from 23.36% to 36.58%. It is interesting to find that the best-fitting ICS in Scenario A was found with a reference volume $V_{\text{Target}}^A/V_{\text{Guzzetti}} = 1.5$. At the same time, it delivered the best fitness ($\Lambda_S = 23.36\%$) among these three scenarios.

The flow paths under the three scenarios were computed in accordance with the ICS-determined source area. The computational domain covered $1275 \times 740$ m$^2$ and the DEM had a resolution of $\Delta x = \Delta y = 5$ m. The simulation period was 101.0153 s, and the flow paths were determined using the moving mass-covered areas of 51 sets of results from 0.0 s to 101.0153 s, with an interval of 2.0203 s. Similar to the illustration of the flow paths in the Hsiaolin event, only the areas covered by flow thickness of more than 10 cm were taken into account when determining the flow paths. Figure 8d–f present the flow paths under the three scenarios. The turquoise color indicates the overlapping area, while the green, orange, and pink areas are covered only by the results computed in Scenarios A, B, and C, respectively. As reported in Table 5, although the discrepancy between the ICS-covered areas $\Lambda_{\text{path}}^{\text{init}}$ was more significant (ranging from 22.16% to 40.12%), the discrepancy for the whole flow paths $\Lambda_{\text{path}}$ was reduced to less than 10%. We suspect that the channelized topography diminished the discrepancy at the early stage.

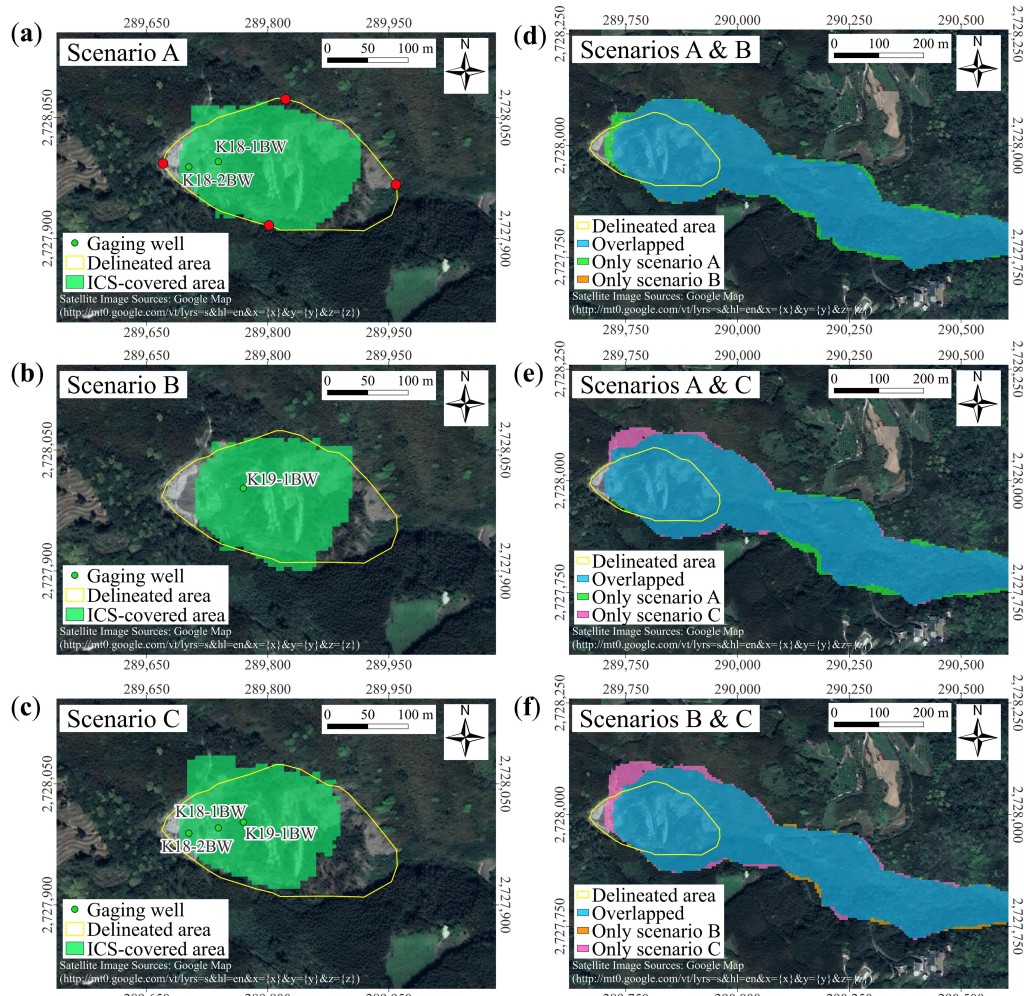

**Figure 8.** The delineated area (outlined by a yellow line), the ICS-covered area, and the consequent flow paths, where the four red points in panel (**a**) are the initial vertices of the GA-ER-ICS search process. The emerald-green indicates the ICS-covered area, the turquoise represents the overlapping area, and the green, orange, and pink regions are covered only by the results computed under Scenarios A, B, and C, respectively: (**a**) ICS-covered area in Scenario A; (**b**) ICS-covered area in Scenario B; (**c**) ICS-covered area in Scenario C; (**d**) Computed flow paths in Scenarios A and B; (**e**) Computed flow paths in Scenarios A and C; and (**f**) Computed flow paths in Scenarios B and C. (Satellite image: http://mt0.google.com/vt/lyrs=s&hl=en&x=x&y=y&z={z} (accessed on 4 May 2022)).

**Table 5.** Comparison of flow paths between various scenarios.

| **Kuanghua-T002** | $\Lambda_{path}^{init}$ | $\Lambda_{path}$ |
|---|---|---|
| Scenarios A vs. B | 22.16% | 6.80% |
| Scenarios A vs. C | 30.42% | 8.93% |
| Scenarios B vs. C | 40.12% | 9.72% |

Among the scenarios, the released mass in Scenario A was 50% more than those in the other two scenarios. Still, the corresponding values of $\Lambda_{path}$ were not as notable as the volume difference. In this regard, the volume of released mass seems not to be the critical factor for the flow paths. On the other hand, the released volumes in Scenarios B and C were identical, but the values of the associated $\Lambda_{path}^{init}$ and $\Lambda_{path}$ were dominant. Nevertheless, the discrepancy in the campaign between Scenarios B and C reduced from 40.12% at the initial stage to 9.72% for the whole flow-flushed region, indicating that the channelized topography in the downstream area plays a significant role in the consequent

flow paths (see also [18]); that is, a considerable discrepancy at the initial stage may be attenuated during the movement in a channelized topography. These findings reveal that the flow paths are not sensitive to various ICSs under a channelized topography. In the three scenarios, the simulated flow paths did not touch the houses sitting in the lower right corner of Figure 8d–f. Although a more detailed study concerning various parameter sets is needed, these results indicate that the buildings do not lie in the center of the flow paths and, so, are at low risk. Such flow path investigations, using constructed ICSs under various scenarios, might provide an excellent representative hazard assessment model for landslide-prone areas.

## 6. Discussion and Concluding Remarks

In the present work, we outlined an efficient methodology integrating a GA approach with the reference ellipse-ICS method, in order to mimic the plausible failure surface, as well as estimate the scarp, for a landslide-prone area. The proposed method does not aim to extract a precise estimation but, instead, to provide a preliminary assessment; especially when no detailed field data are available. The reference ellipse-ICS method [20] utilizes the reference ellipse to construct the ICS. However, the goodness-of-fit to the assigned targets (e.g., shape of the area, failure depths) highly depends on the location and orientation of the reference ellipse. As a matter of course, there are thousands of orientations (candidate ICSs) to be evaluated for a target site, in order to determine the most appropriate ICS. The GA process reduces the number of candidate ICSs by preserving the critical characteristics of the ICS through the concept of evolution, such that one can practically employ a broader range of parameters (treated as genes) in the search process.

For each assigned condition/scenario, the search operation was repeated three times, and the best-fitted one was selected for simulation of the associated flow path. The fitting ability of the constructed ICSs to the main source area of the 2009 Hisiaolin landslide was numerically studied, and the effectiveness of the side-tilting operation and the convergence against the number of generations were examined. It is interesting to note that all of the GA-selected ICSs exhibited similar shapes, with deviation index $\Lambda_S \in [0.2405, 0.3005]$. The results also revealed that the side-tilting operation did not have a remarkable impact on the fitness performance. In addition, it was found that a satisfactory ICS could be found after ten generations among the three runs, although a more stable plateau of fitness was obtained with 15 generations. In the investigation of flow paths, most of the discrepancy took place in the early stage, and the presence of a channelized topography may attenuate the impact of the difference between ICSs on the flow path.

Application of the proposed method to a landslide-prone area (Kuanghua-T002) exhibited a representative example, concerning the utilization of the GA-ER-ICS method for a preliminary hazard assessment, through the delineation of a potential zone without requiring detailed geological structure or hydrological conditions as prerequisites. Based on the records of installed inclinometers, three scenarios (Scenarios A, B, and C) were considered. The GA-ER-ICSs were constructed and selected in accordance with the recorded failure depths in the scenarios. It was found that simultaneously meeting the fitness of the target area, failure depth(s), and the assigned volume of released mass is highly challenging. Hence, some compromises (e.g., the fitness to the delineated area or the landslide volume) may be needed. In the three scenarios, all of the ICS-covered areas deviated from the delineated region, and were distinct from each other. Despite the clear distinction among the ICS-covered zones, simulation of the landslide routes revealed that the consequent flow paths were not sensitive under a channelized topography.

It is worth noting that the ICS is a preliminary approximation to the landslide scarp and should be used for the purpose of scenario investigation by numerical simulation when only limited field data are available. It is intended to make up for a deficiency, instead of replacing conventional slope stability analyses. We should admit that there is inevitably a notable discrepancy between the smooth ICS and the non-trivial natural failure surface. Nevertheless, our investigations support the fact that this discrepancy does

not significantly influence the consequent flow paths (see also [18,20]). The integration of the GA approach with the ER-ICS method has significantly enhanced the efficiency of the searching process. In comparison with the method of exhaustion (ER-ICS), the proposed GA-ER-ICS may reduce the computational time from more than 20 hrs (for ICS-D) to circa 13.5 mins (e.g., for GA-IIb with 15 generations) with a PC (i7-9700 CPU@3.00 GHz×8, 64 GB memory, Linux OS Ubuntu 18.04), when a GPU (NVIDIA GeForce TRX 2080Ti) is utilized. It should be noted that, in the GA-ER-ICS process, a significantly more extensive translation range ($17 \times 17$ grids) is taken into account in the search process, compared to that in the exhaustive method ($7 \times 7$ grids for the ICS-D in [20]). The proposed GA-ER-ICS method and the GPU-accelerated simulation tool [26] facilitate a highly efficient hazard assessment system, which is currently under development. We intend to report updates on the system in due time.

**Author Contributions:** Conceptualization, Y.-C.T. and P.-H.P.; methodology, Y.-C.T.; software, C.-L.W. and C.-J.K.; validation, C.-L.W., H.-K.W. and Y.-C.T.; formal analysis, C.-L.W. and H.-K.W.; investigation, H.-K.W. and Y.-C.T.; resources, Y.-C.T.; data curation, H.-K.W.; writing—original draft preparation, Y.-C.T.; writing—review and editing, Y.-C.T.; visualization, H.-K.W. and P.-H.P.; supervision, Y.-C.T.; project administration, Y.-C.T.; funding acquisition, Y.-C.T. All authors have read and agreed to the published version of the manuscript.

**Funding:** This research was partly founded by the Ministry of Science and Technology, Taiwan (MOST 110-2221-E-006-045-MY2) and the Soil and Water Conservation Bureau, Council of Agriculture, Taiwan (SWCB-111-035).

**Institutional Review Board Statement:** Not applicable.

**Informed Consent Statement:** Not applicable.

**Data Availability Statement:** The data that support the findings of this study are available from the corresponding author upon reasonable request.

**Acknowledgments:** The authors are grateful to Chun-Wei Lu for his support and discussions related to the GA method.

**Conflicts of Interest:** The authors declare no conflict of interest.

## Abbreviations

The following abbreviations are used in this manuscript:

| | |
|---|---|
| CUDA | Compute Unified Device Architecture |
| DEM | Digital Elevation Model |
| ER-ICS | Ellipse-Referenced ICS |
| GA | Genetic Algorithm |
| GA-ER-ICS | Genetic Algorithm-selected Ellipse-Referenced ICS |
| GPU | Graphics Processing Unit |
| ICS | Idealized Curved Surface |
| RE | Reference Ellipse |
| SLBL | Sloping Local Base Level |
| UAV | Unmanned Aircraft System |

## Appendix A

In the GPU-accelerated simulation tool (MoSES_2PDF [26]), there are five material parameters $(\alpha_\rho, \delta_b, C_d, N_R, \vartheta_b)$ and one initial concentration, $\phi_0^s$, for the flow body to be set (cf. Tai et al. [38]). Here, $\alpha_\rho = \rho^f / \rho^s$ is the density of the interstitial fluid to the solid constituent, $\delta_b$ denotes the angle of basal friction of the solid constituent, $C_d$ represents the drag coefficient between the interstitial fluid and the solid constituent, $N_R$ is proportional to the inverse viscosity (similar to the Reynolds number), and $\vartheta_b$ denotes the fluid friction coefficient at the basal surface. It should be noted that the values of the material parameters depend on the composition of the moving mass. The determination of their values requires

further study, and is beyond the current scope of this paper. To enable a comparison with the results provided in previous studies [18,20,38], identical values were adopted in the investigation of the consequent flow paths, which are listed in Table A1.

**Table A1.** Material parameters and initial volume fraction used for computing flow paths.

| Parameter | $\alpha_\rho$ | $\delta_b$ | $c_D$ | $N_R$ | $\vartheta_b$ | $\phi_0^s$ |
| --- | --- | --- | --- | --- | --- | --- |
| Used value | 1.42/2.6 | 16° | 6.0 | 268 | 5.0 | 0.5 |

**Appendix B**

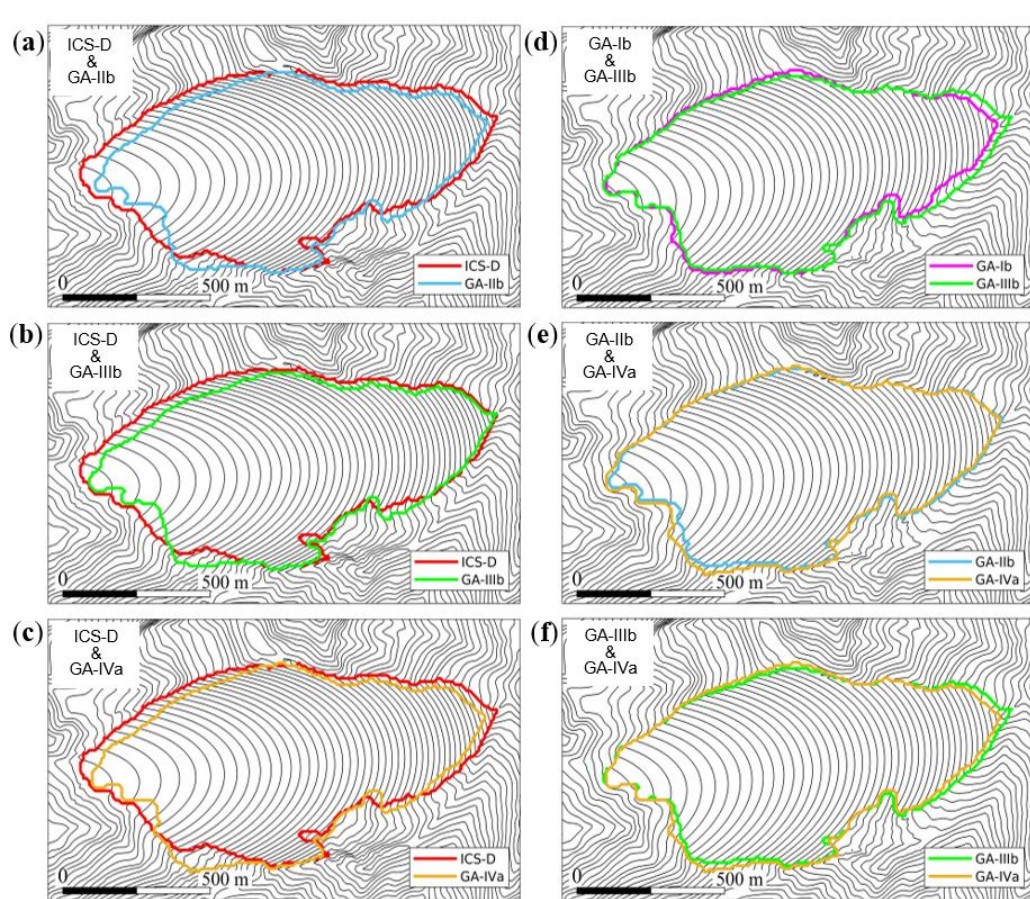

**Figure A1.** Outlines of the ICS-determined failure (source) areas used for computing the flow paths in the corresponding panels in Figure 7.

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
