# Peer review of "An Approach for Preliminary Landslide Scarp Assessment with Genetic Algorithm (GA)"

_water, doi:10.3390/w14152400_

Round 1
Reviewer 1 Report
Hi authors
The paper presented a new and time saving method to map and predict a landslide impact where scanty field data is available. If a detailed flow chart for the entire processes is provided, it will help readers better understand the concepts and steps involved as shown for the GA process.
Since it is a multi-disciplinary article, for a better understanding of all kind of readers, it may be a benefit if input and output data structures are bit more explained.
Adding more pictorial representation of processes or outputs in each process may make it more pleasant reading.
The MS requires a thorough grammatical correction. I have annotated the slight grammatical corrections initially, which are attached herewith for your reference. but the authors should ensure that the entire manuscript should be read by a native English speaker

Reviewer 2 Report
A very interesting development direction in the assessment of the occurrence of a landslide
Author Response
Many thanks for the positive comment! No review report to reply.
Reviewer 3 Report
The article looks at a very interesting problem. It is always crucial to identify slide prone areas and mark plausible areas that the slide could reach out in case of failures. This is especially important for slide-related hazard mitigation and preparedness.
Below are main comments that I think lacks from the article looking from a geotechnical perspective
I see that ER-ICS is proposed as a preliminary assessing tool where the local field data is limited. Still, landslides involve soil materials, and it is important to give better description of the materials/soils at the sites presented in the article. Reference is made to article 20 but it is not appropriate to make a key aspect of this work reliant on another article (one should not have to find article 20 to make this work complete). So, the authors should give as much characterization of materials at the site as possible.
Table A1 gives material parameters symbol and numerical values – it is important to describe the parameters and explain their physical meaning. What do they represent?
To use the proposed tool, one needs to have values for the parameters along with detailed geometrical data. So, the question is that how would one determine these parameters a priori for a given site? Back calculation of run-out of landslides is relatively easy and has been done with various methods over the years. The final landslide scarp gives a control, and this is a big advantage. However, in real live we want to make such calculation in advance of failures. So what is the proposal from authors to get the parameters if one has to use the proposed methodology on a site that needs hazard assessment/disaster mitigation? Any tests or correlations with established soil properties? It is crucial to address this aspect in the article.
What governs the strength versus failure relationship to delineate initial source area?
I assume that the approach adopted is modeling once failure is initiated. The governing physics in that process could be elaborated a bit more. In extremely highly sensitive clays (Easter Canada and Scandinavia) then the materials could behave like a soup after failure. In other cases of debris flow then there is a significant water content that drives the flow of these slides. So, in what context is the method thought to be used. Again, material and its physics need to be elaborated precisely.
Below are few editorial
For fig 5 – the caption for (e) is repeated twice – the last one should have been (f)
Fig 7 – lacks scale
Fig 7 gives flow paths of a historical slide (The 2009 Hsiaolin Landslide). The final landslide scarp should be plotted with dotted lines along the result of the numerical simulation to evaluate the results.
